# Quantitative analysis questions the role of MeCP2 as a global regulator of alternative splicing

Kashyap Chhatbar[1,2], Justyna Cholewa-Waclaw[2], Ruth Shah[2], Adrian Bird[2], Guido Sanguinetti[1,3]*

1 School of Informatics, University of Edinburgh, Edinburgh, United Kingdom, 2 The Wellcome Centre for Cell Biology, University of Edinburgh, Edinburgh, United Kingdom, 3 International School for Advanced Studies (SISSA), Trieste, Italy

* gsanguin@sissa.it

**Data Availability Statement:** The article is based on publicly available data and accession codes are within the manuscript.

## Abstract

MeCP2 is an abundant protein in mature nerve cells, where it binds to DNA sequences containing methylated cytosine. Mutations in the *MECP2* gene cause the severe neurological disorder Rett syndrome (RTT), provoking intensive study of the underlying molecular mechanisms. Multiple functions have been proposed, one of which involves a regulatory role in splicing. Here we leverage the recent availability of high-quality transcriptomic data sets to probe quantitatively the potential influence of MeCP2 on alternative splicing. Using a variety of machine learning approaches that can capture both linear and non-linear associations, we show that widely different levels of MeCP2 have a minimal effect on alternative splicing in three different systems. Alternative splicing was also apparently indifferent to developmental changes in DNA methylation levels. Our results suggest that regulation of splicing is not a major function of MeCP2. They also highlight the importance of multi-variate quantitative analyses in the formulation of biological hypotheses.

## Author summary

Rett Syndrome (RTT) is a devastating neurological disorder affecting approximately 1 in 10,000 female births. Most cases of RTT are caused by mutations in the gene identified as methyl-CG binding protein 2 (*MECP2*) which is an epigenetic reader of DNA methylation. Although the primary function of MeCP2 is to recruit NCoR to methylated sites in the genome, the downstream effect on gene expression is subtle and multiple additional functions have been proposed. Here we focus on the influence of MeCP2 on one of these: alternative splicing to generate different messenger RNAs from a single primary transcript. Using machine learning approaches, we show that neither MeCP2 nor DNA methylation influence alternative splicing. Our results emphasize the importance of multi-variate quantitative analyses and they challenge the over-interpretation of causal relationships based on high-throughput data sets.

**Funding:** Kashyap Chhatbar was supported by a scholarship from College of Science and Engineering, University of Edinburgh. This work was supported by Wellcome Centre grant (091580/Z/10/Z), a Wellcome Investigator Award (107930/Z/15/Z), the Rett Syndrome Research Trust and a European Research Council Advanced grant (EC 694295 Gen-Epix) to Adrian Bird. Adrian Bird is a member of the Simons Initiative for the Developing Brain at the University of Edinburgh. Ruth Shah was supported by a Wellcome Trust 4-y PhD studentship. The funders had no role in study design, data collection and analysis, decision to publish, or preparation of the manuscript.

**Competing interests:** The authors have declared that no competing interests exist.

## Introduction

MeCP2 (methyl CpG binding protein 2) is an important mediator of epigenetic regulation in mammalian cells, and particularly in neurons [1–3]. Mutations that lead to depletion or over-expression of the *MECP2* gene are associated with severe neurological diseases, most notably Rett syndrome, a devastating autism-spectrum disease affecting approximately 1 in 10,000 female births [4]. MeCP2 is a chromatin protein that binds to modified cytosine residues, primarily mCG [5], the most abundant modification in mammalian genomes, but also mCA, particularly mCAC [6], which is a feature of neuronal cells [7, 8]. This association with chromatin enables MeCP2 to interpret epigenetic signals to modulate gene expression events. By far the best documented role of MeCP2 is repression of transcription: several large-scale studies have confirmed this role, and proposed mechanistic models that can account for the effect of MeCP2 on transcription [9–11]. Other roles for MeCP2 have also been hypothesised, however, including chromatin compaction, micro-RNA processing, and regulation of alternative splicing (reviewed in reference [12]). These hypotheses highlight alternative roles of MeCP2 in orchestrating gene expression at transcriptional and post-transcriptional levels, but they have varying degrees of support from data.

In this paper, we focus on the possibility that MeCP2 is a regulator of alternative splicing. This was first proposed after a physical interaction with the splicing factor YB1 was observed [13] in 2005. Subsequent studies provided additional support; most recently via a modest but significant association between changes in DNA methylation and intron retention in the transition between promyelocytes and granulocytes [14]. A role of MeCP2 in splicing is plausible from a mechanistic perspective [9–11] if binding of the protein serves as a "brake" for polymerase, leading to a reduced transcription rate, which in turn can alter splicing preferences [15]. These studies relied on correlative, univariate analyses of data from experiments which were not designed to specifically test the association of MeCP2 with splicing. This renders the conclusions vulnerable to confounding effects, which could explain both changes in splicing and DNA methylation/MeCP2 occupancy. Although a measure of significance of the association (assuming no confounding factors) was deduced, the effect size was not quantified. As a result, questions regarding the extent to which variation in splicing can be attributed to changes in DNA methylation or MeCP2 levels remain open.

Here we revisit the evidence for this phenomenon, leveraging recent, high-quality data sets which track transcriptomic changes in experiments specifically designed to reflect changes in MeCP2 activity. These new data sets offer an unprecedented opportunity to tackle this question statistically, providing a quantitative estimate of MeCP2's effect while carefully controlling for confounding effects.

## Results

It has been suggested that DNA methylation regulates alternative splicing through different mechanisms [16]. We evaluated the relationship from two perspectives: either dependent on MeCP2 or independent of MeCP2. Considering the former first, we sought transcriptomic data sets from cells or tissues with different controlled levels of MeCP2 while keeping DNA methylation constant. Several comprehensive data sets have recently been collected in situations that closely mimic this idealised scenario (Table 1). In particular, our previous work [9] captured transcriptomic data sets in Lund Human Mesencephalic (LUHMES)-derived human dopaminergic neurons that expressed widely different levels of MeCP2, while total levels of DNA methylation remained constant. Additionally, Boxer et al. [17] comprehensively captured transcriptomes and methylomes from wildtype (*WT*) and MeCP2 (*KO*) mouse brain tissues. To investigate the latter, we examined the *Dnmt1/3a/3b* triple-knockout mouse

**Table 1. Sequencing data sets considered for multi-variate quantitative analysis.**

| Accession | Source | Total RNA-seq Coverage | Reps | Type |
|---|---|---|---|---|
| GSE125660 [9] | MeCP2 *WT* Neurons | $218.2 \times 10^6$ pairs | 4 | BS-seq, RNA-seq, ChIP-seq |
| | MeCP2 *KO* Neurons | $214.2 \times 10^6$ pairs | 4 | RNA-seq, ChIP-seq |
| | MeCP2 *OE4x* Neurons | $217.5 \times 10^6$ pairs | 4 | |
| | MeCP2 *OE11x* Neurons | $220.4 \times 10^6$ pairs | 4 | |
| GSE128186 [17] | *WT* brain tissue | $713.9 \times 10^6$ reads | 10 | BS-seq, RNA-seq |
| | MeCP2 *KO* brain tissue | $796.5 \times 10^6$ reads | 10 | |
| GSE64910 [18] | *WT* mESCs | $49.1 \times 10^6$ reads | 2 | BS-seq, RNA-seq |
| | *DNMT-TKO* mESCs | $50.3 \times 10^6$ reads | 2 | |
| GSE67867 [19] | *WT* mESCs | $357.3 \times 10^6$ reads | 3 | RNA-seq |
| | *DNMT-TKO* mESCs | $364.8 \times 10^6$ reads | 3 | |
| GSE103214 [20] | *Pv* Neurons (1 week) | $37.7 \times 10^6$ reads | 2 | BS-seq, RNA-seq |
| | *Pv* Neurons (3 weeks) | $27.6 \times 10^6$ reads | 1 | |
| | *Pv* Neurons (8 weeks) | $97.5 \times 10^6$ reads | 2 | |
| | *Vip* Neurons (1 week) | $17.9 \times 10^6$ reads | 1 | |
| | *Vip* Neurons (3 weeks) | $57.9 \times 10^6$ reads | 2 | |
| | *Vip* Neurons (8 weeks) | $45.3 \times 10^6$ reads | 2 | |
| GSE48307 [23] GSE85517 [14] | Promyelocytes | $63.5 \times 10^6$ reads | 1 | BS-seq, RNA-seq, ChIP-seq |
| | Granulocytes | $59.7 \times 10^6$ reads | 1 | |

embryonic stem cells (*DNMT-TKO* mESCs) that lack DNA methyltransferase activity [18, 19]. Yearim et al. performed splicing analyses on a relatively low-coverage data set and suggested that DNA methylation may influence splicing of alternate exons [18]. Therefore, we sought to revisit this conclusion using a more recent, higher coverage *DNMT-TKO* dataset [19] to investigate the role of DNA methylation in influencing alternative splicing independent of MeCP2. Finally, it is also possible that splicing regulation is due to a combination of changes in MeCP2 levels and DNA methylation. In early postnatal brain, the level of genomic mCA, a major determinant of MeCP2 binding, concurrently increases with the level of MeCP2 [5, 20, 21]. Stroud et al. [20] profiled the transcriptomes and methylomes of differentiating mouse neurons in the first few weeks after birth. Therefore, developing neurons can be considered an ideal scenario where one can investigate the combined effects of MeCP2 and DNA methylation. For completeness, we also re-analysed the non-neuronal data sets from promyelocytes and granulocytes [14] which first led to suggest the potential link between MeCP2, DNA methylation and alternative splicing. MeCP2 RNA expression levels show predominant expression in neurons and brain tissue while mESCs and granulocyte express significantly lower levels (S1 Fig). All of these data sets provide a good level of replication and high sequencing depth, with the exception of the early developmental time points of [20], where experimental constraints limited the achievable depth.

Adequate sequencing coverage is important for robust alternative splicing analysis, and some of the data sets we employ, particularly the early developmental times in the Stroud et al. data [20], have relatively low coverage. To address this, we use a recently proposed splicing quantification strategy, BRIE (Bayesian Regression for Isoform Estimation [22]), which uses Bayesian sequence-derived, informative priors to provide robust estimates of splicing ratios even at very low coverage levels such as encountered in single cell RNA-seq data.

Summary statistics for all the data sets used, including sequencing depth and replication level, are provided in Table 1.

## Differential splicing analysis questions the function of MeCP2 and DNA methylation as a global regulator of alternative splicing

To explore the effects of MeCP2 on splicing and gene expression, we initially performed differential splicing and differential gene expression analyses for a number of data sets which investigated changes in MeCP2 levels (Table 1). Varying the abundance of MeCP2 in cultured human neurons over a wide range (absence (*KO*), 1x (*WT*), 4x and 11x wildtype (*OE*) levels) is accompanied by many changes in gene expression relative to wildtype (*KO* = 178; *4xOE* = 2548; *11xOE* = 2348 at a significance threshold of p-adjusted value < 0.05), but relatively few differential splicing events (*KO* = 101; *4xOE* = 120; *11xOE* = 73 at a significance threshold of Bayes factor $\geq$ 3) (S2A and S2B Fig). We observe lower number of differential splicing events in *KO* mouse brain (57) compared to changes in gene expression (2943) (S2C and S2D Fig). This suggests that changing the levels of MeCP2 (*in vitro* or *in vivo*) results in sizable changes in transcription but negligible changes in splicing.

In *WT* mouse neurons, where levels of DNA methylation and MeCP2 increase during development, we found that the number of differential splicing events (489, 408) is lower compared to the number of differential gene expression events (1122, 538) when 8 weeks old and 1 week old Parvalbumin (*Pv*) and Vasoactive intestinal peptide (*Vip*) neurons are compared (S2E and S2F Fig). Comparisons between 8 weeks old and 3 weeks old *Pv* and *Vip* neurons showed more differential splicing events (60, 527) than differentially regulated genes (4, 118) (S2E and S2F Fig). Based on this analysis, we could not establish whether MeCP2 and DNA methylation predominantly perturbs transcription or alternative splicing in developing neurons. Because of the increase in CA methylation [20, 21] and the dynamic nature of alternative splicing [24] in brain development, it is necessary to investigate the role of DNA methylation independent of developmental effects and MeCP2. Therefore, we performed similar analysis for *DNMT-TKO* mESCs in comparison to *WT* mESCs. In the absence of DNA methylation, many genes are differentially regulated (7342), but there are relatively few differential splicing events (134) (S2G and S2H Fig). Based on this analysis, DNA methylation predominantly perturbs transcription, with a much smaller effect on splicing. In a separate data set for *DNMT-TKO* mESCs [18], the number of differentially regulated genes (102) was only marginally higher than number of differential splicing events (71) (S2I and S2J Fig). Differential gene expression analysis was not possible on the RNA-seq data set of Promyelocytes and Granulocytes [14, 23] due to the absence of replicates. The lower number of differential splicing events compared to differentially expressed genes is consistent at different significance thresholds (S2 Fig) in all comparisons except in developing neurons. To investigate whether differential splicing events are directly associated with context specific DNA methylation or MeCP2 binding, we performed correlation analysis on significantctly altered events (S2B, S2D, S2F, S2H and S2J Fig). We found that there is no apparent linear relationship between changes in splicing and changes in DNA methylation or MeCP2 binding (S1 Table) except a modest one for *Pv* neurons (highlighted in S1 Table). By performing classical differential gene expression and differential splicing analysis, we find that MeCP2 primarily functions as a regulator of gene expression and has minimal effect on splicing. Whether the primary role of DNA methylation and neuronal development is in influencing gene expression or alternative splicing, is uncertain. To test this in a more quantitative manner, we used machine learning techniques in our subsequent multivariate analysis.

## Sequence features are highly predictive of splicing ratios

Machine learning approaches have previously shown that genomic sequence features are highly predictive of exon inclusion/exclusion ratios (splicing ratios) and the splicing code

assembled using these features helped characterise alternative splicing on a genome-wide scale [25, 26]. The splicing ratios were calculated using several independent methods, and our analysis indicated a strong correlation between results obtained with different methods (S3A and S3E Fig). Methods that directly model splicing ratios such as Miso [27] and BRIE [22] showed particularly strong agreement (S3A Fig). To quantify the explanatory power of sequence features in predicting splicing ratios in the current data sets, we used two independent multivariate regression models: Linear Regression and Random Forest Regression, the latter being a flexible non-parametric approach. We first validated that the relationship between the splicing code and splicing ratios [25, 26] could be confirmed for all the data sets described in Table 1 We find that both models explain 50%-65% of the variation (Fig 1A, 1B, 1D, 1E and 1F left panels). This effect is quantified by the metric fraction of variation explained (FVE) which is analogous to $R^2$. When using splicing ratios from MISO, we still retain 35%-50% of the explanatory power of sequence-derived features (Fig 1A, 1B, 1D, 1E and 1F right panels). We note that predictive models achieved lower accuracy in *Vip* neurons and *Pv* neurons during early mouse brain development (Fig 1C), possibly due to the highly dynamic nature of splicing in early development [24]. We confirm that splicing ratios can be accurately regressed against sequence-derived features and this analysis provided positive controls for further analysis.

## MeCP2 changes account for a negligible fraction of splicing variation

We then probed whether changes in inclusion/exclusion ratios between different cell types could be explained by changing levels of MeCP2. We refer to differential exon inclusion/exclusion between different cell types as splicing variation (not to be confused with splicing ratios, which are the ratio of inclusive/exclusive isoform abundance within the same sample). Since global changes in transcription correlate with MeCP2 levels [9] and sequence-derived features can predict splicing ratios (Fig 1) [25, 26], our hypothesis is that any role of MeCP2 in regulating splicing should be reflected in improved predictions of differential exon inclusion/exclusion. Since global DNA methylation remains constant in neurons expressing multiple levels of MeCP2 (S4A Fig), we assume that DNA methylation at exons and introns also remain constant. To allow for the possibility that MeCP2 regulation of splicing might be dependent on the local sequence context, we used regression models with both MeCP2 levels and sequence features as input variables.

When we quantified the explanatory power of contrasting MeCP2 levels (*in vitro*) by regressing splicing variation against sequence features and MeCP2 binding, we could only explain 0-3% of the splicing variation regardless of the quantification method or model (Fig 2A left panel). In some instances, the model fit is extremely poor, as evidenced by negative levels of splicing variation explained (Fig 2A right panel). To recapitulate a similar situation *in vivo*, we used data sets from [17] which profiled transcriptomes and methylomes of *WT* and MeCP2 *KO* mouse brain tissues at very high sequencing depth. We found that DNA methylation levels, quantified from WGBS-seq, are statistically indistinguishable between *WT* and MeCP2 *KO* mice brains (S4B and S4C Fig): correlations between methylomes of different WT replicates are identical to correlations between methylomes in *WT* and *KO* samples. This further supports our assumption that DNA methylation remains constant in neurons expressing different levels of MeCP2. In mouse brain data sets, when we quantified the explanatory power of MeCP2 by regressing splicing variation against sequence features, the model fit was once again poor, with negligible or even negative levels of splicing variation explained (Fig 2B). Importantly, the 57 differentially spliced genes in this data set did not show any significant level of differential methylation (S1 Table), ruling out a potentially confounding effect of a compensatory local change in methylation. Therefore, multiple levels of MeCP2 (*in vitro* and

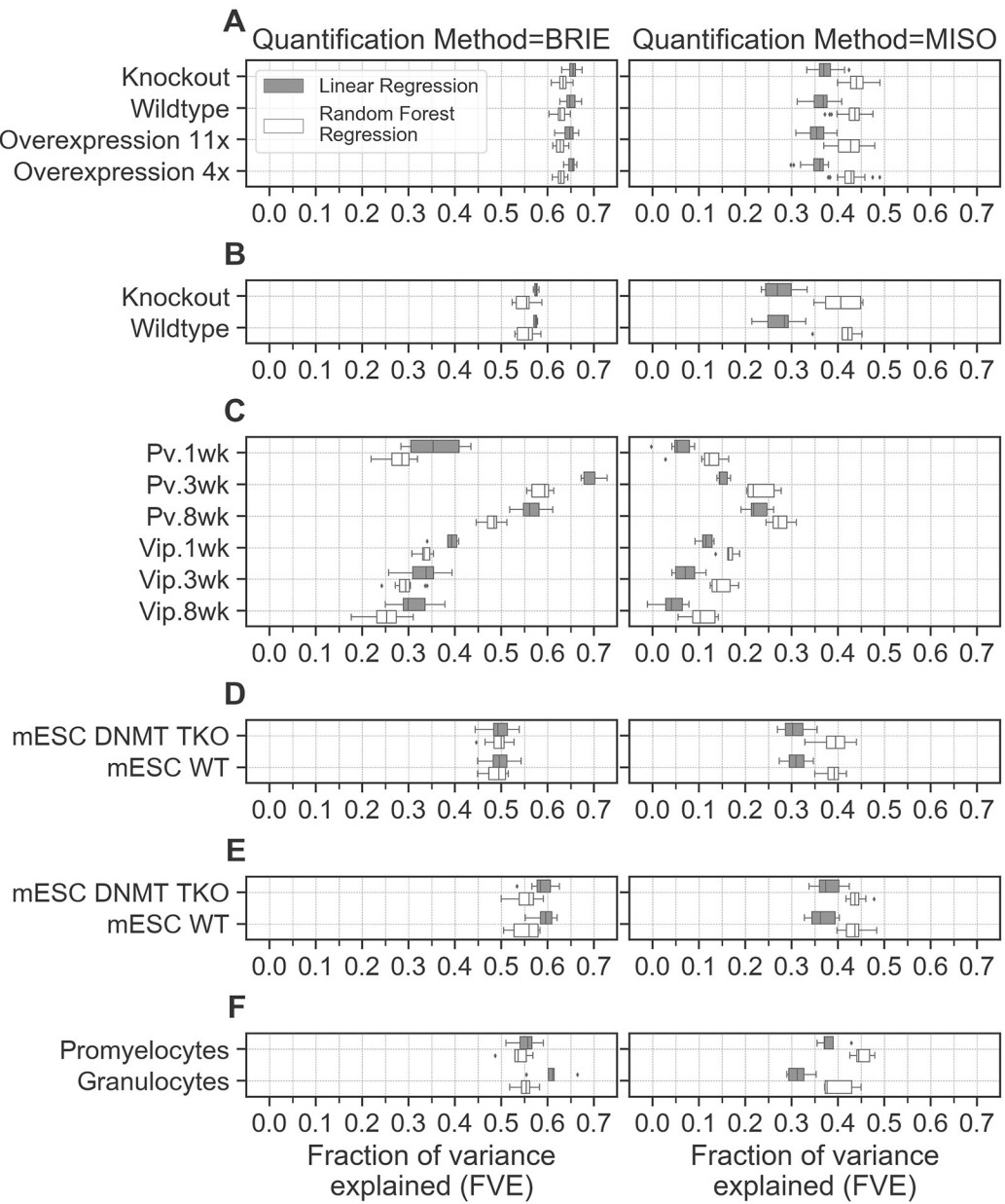

**Fig 1. Fraction of variance explained by regressing sequence features against splicing ratios. (A)** Cultured human neurons expressing multiple levels of MeCP2 [9] **(B)** Wildtype (*WT*) and MeCP2 Knockout (*KO*) brain tissue [17] **(C)** *Pv* and *Vip* neurons at 1 week, 3 weeks and 8 weeks [20] **(D)** *DNMT-TKO* and *WT* mESCs [18] **(E)** *DNMT-TKO* and *WT* mESCs [19] **(F)** Promyelocytes and Granulocytes [14, 23]. See Table 1 for detailed information about data sets.

*in vivo*) combined with sequence features cannot predict splicing variation, suggesting that MeCP2 does not regulate splicing, even when sequence context is taken into account.

## DNA methylation changes account for a negligible fraction of splicing variation

MeCP2 binding in neurons is strongly associated with DNA methylation in a CG and CA context [6]. We asked whether changes in context specific DNA methylation in developing brain,

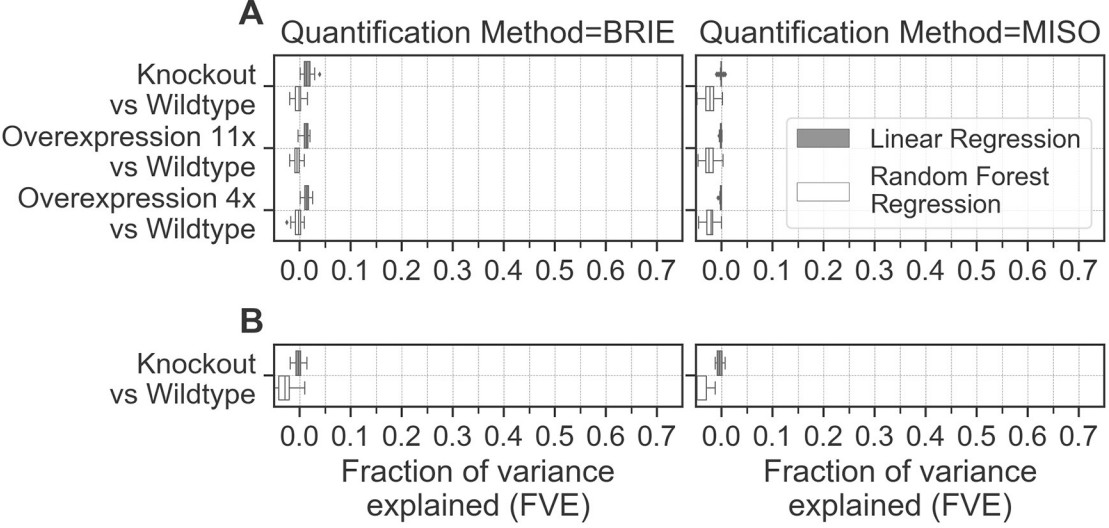

**Fig 2. Explanatory power of varying MeCP2 levels (*in vitro* and *in vivo*). (A)** Fraction of variance explained by regressing splicing variation against sequence features combined with MeCP2 binding in cultured human neurons [9]. **(B)** Fraction of variance explained by regressing splicing variation against sequence features combined with changes in DNA methylation in *WT* and MeCP2 *KO* mouse brain [17].

the primary determinant of MeCP2 binding, affects splicing variation. To do so, we calculated the changes in DNA methylation at alternative exons and surrounding regions for both dinucleotides CG and CA (Fig A in S1 Appendix) in developing mouse neurons [20]. In this instance, splicing variation is the difference in splicing ratio across *Pv* and *Vip* neurons at different ages. We quantified the explanatory power of methylation features by regressing splicing variation against features derived from DNA methylation, as well as sequence features. The fraction of variance explained by the models is 0-10% (Fig 3A left panel). The analysis fails to capture any relationship (linear/non-linear) when splicing variation is regressed against DNA methylation alone (S5A Fig). Notice, however, that the inclusion of sequence features accounts for context-specific effects of DNA methylation, in a scenario where MeCP2 is essential for normal function of nerve cells. Therefore, context specific DNA methylation and sequence features in developing neurons are only at best modest predictors of splicing variation.

We then specifically set out to test whether changes in DNA methylation and their sequence context can predict splicing variation independently of MeCP2. To investigate the relationship between DNA methylation and alternative splicing independent of MeCP2 and developmental effects, we chose *DNMT-TKO* mESCs which lack DNA methylation [18, 19]. As these cells also express low levels of MeCP2 (S1 Fig), this analysis is important to rule out any confounding effects on our conclusions possibly caused by the high correlation between MeCP2 binding and DNA methylation in neuronal cells. Here, splicing variation is the difference in splicing ratio between *DNMT-TKO* mESCs and *WT* mESCs. We quantified the explanatory power of methylation features by regressing splicing variation against DNA methylation and associated DNA sequence features. Both regression models are able to account for a minimal fraction (0-10%) of the observed variance in splicing ratios (Fig 3C left panel). In a separate *DNMT-TKO* mESCs data set, the accuracy was lower (0-5%) (Fig 3B left panel). Therefore, DNA methylation when considered in a scenario independent of MeCP2 and developmental effects cannot predict splicing variation.

The promyelocytes-granulocytes data set is the only one where methylation levels combined with sequence features do appear to explain a small fraction of variance in splicing

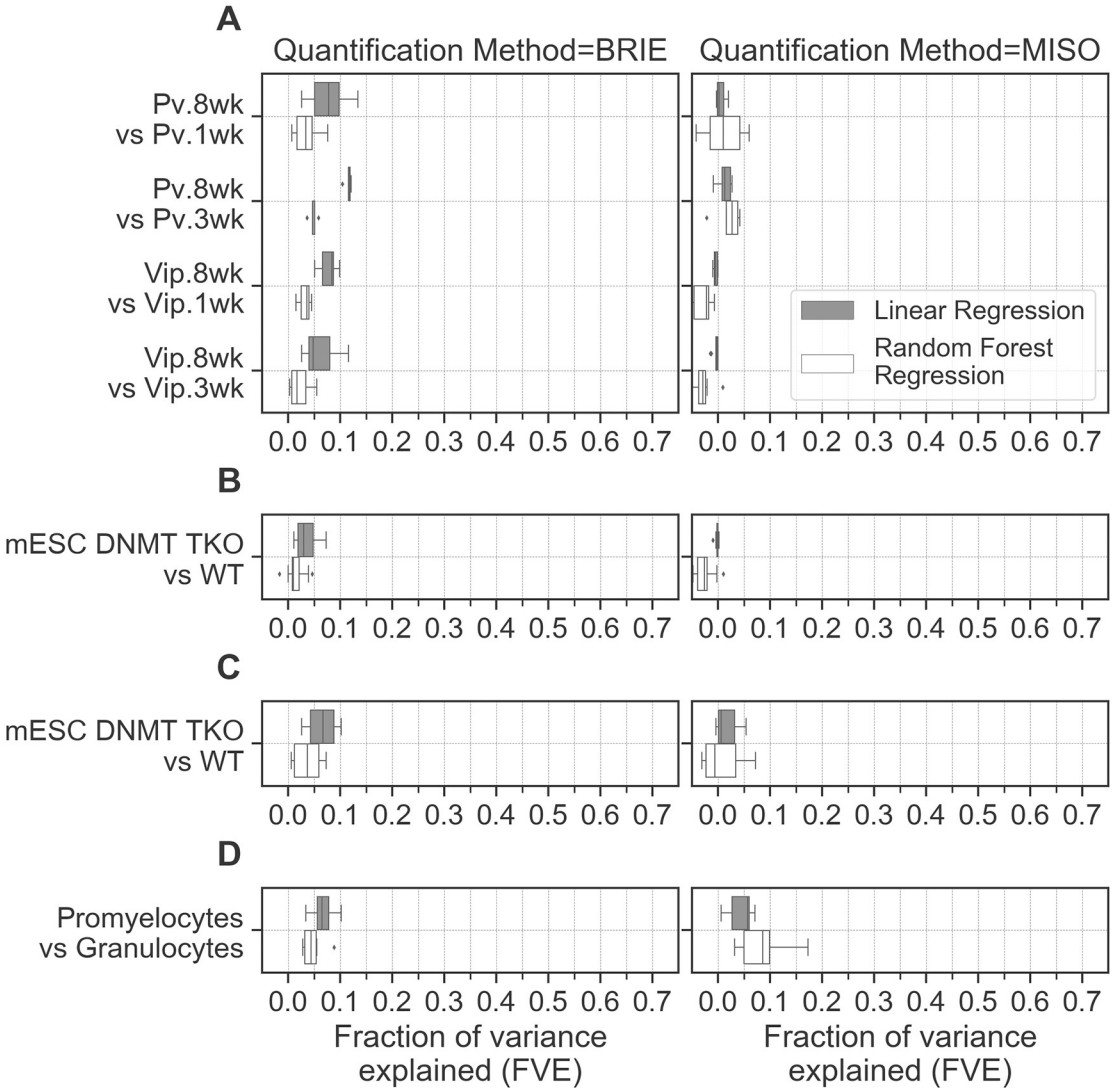

**Fig 3. Fraction of variance explained by regressing splicing variation against sequence features along with DNA methylation features. (A)** *Pv* and *Vip* neurons at 1 week, 3 weeks and 8 weeks [20]**(B)** *DNMT-TKO* and *WT* mESCs [19]**(C)** *DNMT-TKO* and *WT* mESCs [18]**(D)** Promyelocytes and Granulocytes [14, 23].

variation (5-10%) (Fig 3D). However, even in this case, the fraction of variance explained remains modest when compared to the explanatory power of sequence features in predicting splicing ratios (65% at best and 25% at worst, Fig 1 left panels). It is important to stress that both regression models perform worse when DNA methyation features alone are used to predict splicing variation (S3B Fig), rendering the initial observation moot [14]. In addition to the absence of replicates, as noted above, a further reason for exercising caution is that this study did not explicitly control for changes in MeCP2 abundance.

## Discussion

MeCP2 is a major interpreter of the cell's epigenetic state whose effects on transcription are extensively categorised and recognised as consistent in multiple systems [2, 6, 9–11]. In addition to its effects on gene expression, multiple other potential functions for MeCP2 have been

proposed. However, its high abundance in neurons and pervasive presence in the chromatin environment create difficulties in attributing functions to MeCP2 based purely on observational studies. One often cited function of DNA methylation and its cognate interpreter, MeCP2, is regulation of alternative splicing. Here we have used multivariate analysis of recent data sets to examine this hypothesis in systems where either DNA methylation levels or MeCP2 levels are varied either alone or in combination. The results reveal that MeCP2's role as a global regulator of RNA splicing is minimal and therefore of questionable biological significance.

While statistically our conclusion is robust, it must be qualified in several ways. First of all, we focused on detecting a global role in splicing for MeCP2. The fact that MeCP2 variation explains a negligible fraction of global splicing variation does not imply that MeCP2 plays no role in regulating splicing for specific transcripts. Our analysis would not detect a role of this kind, which would require more targeted experiments to characterise and validate any affected genes. It should however be pointed out that our analysis of splicing variation of *WT* and MeCP2 *KO* mouse brains highlighted that changes in splicing were not associated with any significant changes in DNA methylation at the same loci, suggesting that even a gene-specific role of MeCP2/DNA methylation in regulating splicing is likely to be complex and/ or indirect. A second important reservation is that the small splicing effects that we did detect are not necessarily unimportant. Diseases associated with MECP2 mutations are long-term pathologies, indicating that cells can survive for years in the presence of MeCP2 aberrations. Therefore, it is likely that the molecular effects of MeCP2 will be subtle, a point well-made previously [2, 6, 9–11]. Thirdly, it should be noted that our best-controlled data sets and most of our analyses apply to neuronal cells where MeCP2 is both abundant and strongly implicated in neuronal cell physiology. While it is appropriate that questions relating to MeCP2 function should focus on neuronal data sets, this might overlook effects in other cell types. Indeed, we did detect a modest statistical association between combined DNA methylation and DNA sequence features in the promyelocyte-granulocyte transition, although there was no relationship between DNA methylation alone and alternative splicing in this system. MeCP2 was initially implicated in regulation of alternative splicing in this myeloid lineage, but genetic depletion of MeCP2 specifically in non-brain tissues has no obvious phenotypic consequences, indicating that the protein is dispensable outside the central nervous system [28].

On a more general note, our analysis sounds a note of caution regarding the risks of over-interpretation of experiments that rely on high-throughput data. While such experiments are key in expanding our understanding of molecular physiology, univariate analyses can mask confounding factors. Hence, we argue that biological hypotheses arising from observational studies should always be further validated by carefully designed experiments and supported by multivariate statistical analyses.

## Materials and methods

### Sequencing data sets

Table 1 details the data sets used in the analysis. These include RNA sequencing (RNA-seq) and Bisulfite sequencing (BS-seq) libraries quantifying DNA methylation levels in *Dnmt1/3a/3b* triple-knockout (*DNMT-TKO*) and wildtype (*WT*) mouse embryonic stem cells (mESCs). Also, data sets concerning MeCP2 include RNA-seq and Chromatin immunoprecipitation followed by sequencing (ChIP-seq) libraries from post-mitotic (day 9 of differentiation) human neurons expressing multiple levels of MeCP2 and BS-seq libraries quantifying DNA methylation patterns from wildtype MeCP2-expressing neurons; RNA-Seq and BS-Seq libraries from mouse neuronal subtypes paravalbumin (*Pv-*) and vasoactive-intestinal-peptide (*Vip-*)

expressing interneurons in mouse brain at different post-natal ages; RNA-seq and BS-seq libraries from cells of the mouse myeloid lineage, namely Granulocytes and Promyelocytes. Information about genome assemblies, annotation, software versions and command-line arguments are detailed in S1 Appendix.

## Quantification of splicing ratios

RNA-Seq data sets are downloaded and sequencing reads are mapped to the reference genome using HISAT2 [29] aligner. Splicing ratios are quantified using Miso [27], BRIE [22] and Cufflinks [30]. We found Cufflinks to perform poorly in simulated data with known $\Psi$ values (S3D Fig) when the sequence reads were single-end (S1 Appendix). The Pearson correlation between Cufflinks and Miso quantification in our data sets reflects this (representative scatter plot S3E Fig). Because two of the data sets that we have considered are sequenced with single-end reads and the resultant suboptimal modelling (S3F Fig), we decided to omit Cufflinks quantifications and regression models from further analyses.

## Differential gene expression

Gene counts are extracted using featureCounts [31] from alignment files. Subsequently, differential gene expression is estimated using DESeq2 [32].

## Differential splicing

Differential splicing is estimated using Miso [27].

## Quantification of DNA methylation patterns

Processed BS-Seq data sets described in Table 1 are downloaded and DNA methylation ratio ($mC$ basecalls to the count of all reads $C$ as $\hat{m}_C = \frac{mC}{C}$) at individual nucleotide for CG and CA dinucleotides is accumulated as BigWig [33] tracks. DNA methylation is calculated in two metrics: 1) Region-specific mean methylation defined as $\hat{m}_C = \frac{\sum m_C}{N_C}$ where $N_C$ is the number of C's within the region 2) Region-specific methylation density defined as $\hat{M}_C = \frac{N_C \cdot \hat{m}_C}{L}$ where $L$ is the length of the region. These two metrics are calculated for all the genes such that every gene has separate quantifications for CA and CG dinucleotides. For each gene, several specific regions are considered 1) ASE (alternative spliced exon) 2) combination of all exons containing the ASE 3) combination of all exons without the ASE 4) combination of all introns 5) combination of all introns without the ASE. A visual representation of the specific regions is described in S1 Appendix.

## Quantification of ChIP-seq signal

Processed ChIP-seq data sets described in Table 1 are downloaded and $log_2 \frac{ChIP}{Input}$ signal is calculated using `bigWigAverageOverBed` [33] for regions around the alternative exon. A visual representation of the specific regions is described in S1 Appendix.

## Regressing splicing ratios with sequence features

First, we use an inverse `probit` transformation of our splicing ratios $\Psi$ i.e, $y = \Phi^{-1}(\Psi)$ to map splicing ratios from the interval [0, 1] to the whole of the real numbers. Subsequently, we model the transformed splicing ratio $y$ as a function of a set of $j$ covariates (sequence features) $X \in \mathbb{R}^j$. We split the data, fit the model and compute the coefficient of determination (FVE) which is analogous to $R^2$, 5 consecutive times (with different splits each time) to avoid

overfitting. The *FVE* calculated is the amount of variation in the test set that can be explained by the covariates (sequence features). $FVE(y, \hat{y}) = 1 - \frac{\sum_{i=0}^{n}(y_i - \hat{y}_i)^2}{\sum_{i=0}^{n}(y_i - \bar{y})^2}$ where $\bar{y} = \frac{1}{n}\sum_{i=0}^{n} y_i$. This is applied for both regression approaches.

**Linear regression.**    We model the *y* (transformed splicing ratio) as a linear function of *X* (sequence features). To regularize the weight and noise parameters in the regression problem, we follow the regularization approach [34, 35] which is implemented in the `scikit-learn` [36] package. Information about arguments and parameters are detailed in S1 Appendix.

**Random forest regression.**    We generate random decision trees that are fitted on sets of training data from *X* (sequence features) and *y* (transformed splicing ratio) to form a meta estimator known as random forests [37], implemented in `scikit-learn` [36] package. Using averaging from a number of trees, we predict the transformed splicing ratio *y* on the test data.

## Regressing differential splicing ratios with differential DNA methylation and sequence features

As discussed in the previous section, we use the same approach for the modelling. In this instance, we add more covariates in the form of DNA methylation features. For the mean methylation metric, we use an inverse `probit` transformation and take the difference.

$$\Delta\Phi^{-1}(\hat{m}_C) = \Phi^{-1}(\hat{m}_{C_t}) - \Phi^{-1}(\hat{m}_{C_c})$$

where *t* = *treatment*/*time* − *point* and *c* = *control*/*time* − *point*. For the methylation density metric, we calculate the difference directly.

$$\Delta\hat{M}_C = \hat{M}_{C_t} - \hat{M}_{C_c}$$

We calculate the differential splicing ratio i.e, $\Delta y = \Delta\Phi^{-1}(\Psi)$

$$\Delta\Phi^{-1}(\Psi) = \Phi^{-1}(\Psi_t) - \Phi^{-1}(\Psi_c)$$

and model it to a set of covariates which now consists of sequence features and DNA methylation feautures.

## Supporting information

**S1 Table. Pearson (*r*) correlation between changes in splicing ratios and changes in DNA methylation for all data sets.**
(PDF)

**S1 Fig. MeCP2 expression across data sets considered for analysis (refer Table 1) as counts per million quantified from RNA-seq data.**
(TIF)

**S2 Fig. Number of differential gene expression and splicing events across data sets. (A)** Number of differentially regulated genes and **(B)** Number of differential splicing events in cultured neurons expressing multiple levels of MeCP2 [9] **(C)** Number of differentially regulated genes and **(D)** Number of differential splicing events in MeCP2 *KO* mouse brains compared to MeCP2 *WT* [17] **(E)** Number of differentially regulated genes and **(F)** Number of differential splicing events in developing mouse neurons [20] **(G)** Number of differentially regulated genes and **(H)** Number of differential splicing events in *DNMT-TKO* mESCs [19] **(I)** Number of differentially regulated genes and **(J)** Number of differential splicing events in *DNMT-TKO*

mESCs [18].
(TIF)

**S3 Fig. Modelling transformed splicing ratios $y = \Phi^{-1}(\Psi)$, estimated using different quantification methods, against sequence features $X$. (A)** Representative example of correlation between quantification methods BRIE and Miso. **(B)** Representative scatter plot showing prediction of splicing ratios using linear regression model learned from BRIE's quantification and sequence features. **(C)** Representative scatter plot showing prediction of splicing ratios using linear regression model learned from Miso's quantification and sequence features. **(D)** Comparison of splicing ratio estimation of known $\Psi$ from simulated data at different coverage. **(E)** Representative example of correlation between quantification methods Cufflinks and Miso. **(F)** Representative scatter plot showing prediction of splicing ratios using linear regression model learned from Cufflinks' quantification and sequence features.
(TIF)

**S4 Fig. DNA methylation levels in neurons expressing multiple levels of MeCP2. (A)** Referencing results from our previous work [9]. High Performance Liquid Chromatography (HPLC) quantification of methylated cytosines in neurons expressing wild-type, 4 times and 11 times MeCP2. **(B)** CG and **(C)** CA methylation across introns quantified from bisulfite sequencing of *WT* and MeCP2 *KO* mouse brains (GSE128172 [17]).
(TIF)

**S5 Fig. Fraction of variance explained by regressing DNA methylation features against differential splicing ratios in (A)** developing mouse neurons [20] **(B)** Promyelocytes and Granulocytes [14, 23].
(TIF)

**S1 Appendix. Detailed information about software, software versions and command line arguments used for analysis.**
(PDF)

## Acknowledgments

We are grateful to Gabriele Schweikert and Duncan Sproul for their valuable input. We also thank Shaun Webb for bioinformatics and technical assistance.

## Author Contributions

**Conceptualization:** Kashyap Chhatbar, Guido Sanguinetti.

**Data curation:** Kashyap Chhatbar.

**Formal analysis:** Kashyap Chhatbar.

**Funding acquisition:** Adrian Bird, Guido Sanguinetti.

**Investigation:** Kashyap Chhatbar, Guido Sanguinetti.

**Methodology:** Kashyap Chhatbar, Justyna Cholewa-Waclaw, Ruth Shah.

**Project administration:** Guido Sanguinetti.

**Resources:** Justyna Cholewa-Waclaw, Ruth Shah.

**Software:** Kashyap Chhatbar.

**Supervision:** Adrian Bird, Guido Sanguinetti.

**Validation:** Kashyap Chhatbar, Justyna Cholewa-Waclaw, Ruth Shah.

**Visualization:** Kashyap Chhatbar.

**Writing – original draft:** Kashyap Chhatbar, Guido Sanguinetti.

**Writing – review & editing:** Kashyap Chhatbar, Adrian Bird, Guido Sanguinetti.

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
