## [Decision Letter · Decision Letter 0]

13 Jul 2020

Dear Dr Sanguinetti,

Thank you very much for submitting your Research Article entitled 'Quantitative analysis questions the role of MeCP2 in alternative splicing' to PLOS Genetics. Your manuscript was fully evaluated at the editorial level and by independent peer reviewers. The reviewers appreciated the attention to an important problem, but raised some concerns about the current manuscript. Based on the reviews, we will not be able to accept this version of the manuscript, but we would be willing to review again a much-revised version. We cannot, of course, promise publication at that time.

If you decide to revise the manuscript for further consideration at PLOS Genetics, please aim to resubmit within the next 60 days, unless it will take extra time to address the concerns of the reviewers, in which case we would appreciate an expected resubmission date by email to plosgenetics@plos.org.

[LINK]

We are sorry that we cannot be more positive about your manuscript at this stage. Please do not hesitate to contact us if you have any concerns or questions.

Yours sincerely,

Dirk Schübeler

Associate Editor

PLOS Genetics

John Greally

Section Editor: Epigenetics

PLOS Genetics

Reviewer's Responses to Questions

**Comments to the Authors:**

Reviewer #1: In this study, Chhatbar and colleagues investigate the role of MeCP2 in regulating alternative splicing. MeCP2 is a methyl-DNA binding protein, and over the last decades multiple functions have ben assigned to this protein, including regulation of alternative splicing. The implication of MeCP2 in splicing regulation was mainly based on correlative analysis. In this study, the authors utilise new datasets to identify changes in alternative splicing introduced by MeCP2 and DNA methylation. This is an important contribution towards understanding MeCP2 biology and its contribution to neuronal function. However, there are a few points that are not completely clear.

First, the authors investigate the contribution of MeCP2 to splicing through analysis of transcriptomic data from cells with different levels of MECP2 but constant total methylation data. They find that MeCP2 primarily functions as a regulator of gene expression and has minimal effect on splicing. In these comparisons, the authors mention that global DNA methylation is constant, however there is no analysis provided that supports that conclusion. Differential methylation could be restricted to a small number of regions and the authors need to exclude that the alternative splicing events (even if low in number) are not occurring at these differentially methylated sites. The authors utilise DNMT-TKO ES cells to address how much differential methylation would influence splicing, however there could be profound differences between ES cells and neurons in DNA-methylation-mediated splicing regulation, and this needs to be addressed.

Next they apply machine learning to identify the contribution of DNA methylation and MECP2 to alternative splicing. They first show that they retrieve a good prediction based on DNA sequence features, which serves as positive control. Finally they show that neither MeCP2 or DNA methylation allow prediction of splicing events, globally. This assumes that, if methylation or MeCP2 would contribute to splicing, this effect should be visible for all transcripts or exon/intron boundaries throughout the genome. This also assumes that DNA methylation and binding of MeCP2 is constant across the entire genome and over all exon/intron boundaries - which is not the case. Therefore, a context-dependent role can not be excluded where MeCP2 activity or DNA methylation would influence splicing only if certain criteria are met. E.g. only a at few splicing sites where high MeCP2 binding or high DNA methylation levels are present to influence splicing. I am not sure if the model applied by the authors fully takes these variations in consideration.

Reviewer #2: MeCP2 is an abundant meCpA and meCpG-binding protein with essential roles in neurons. In humans, its mutations cause Rett syndrome. Determining the molecular basis of the disease has been challenging. A role in transcription repression now seems established, but this does not rule out other functions. In particular, as DNA methylation is linked to splicing control, the possibility that MeCP2 regulates splicing has been raised in several publications. Most recently, Wong et al showed in a 2017 Nat Comm paper (PMID 28480880) that MeCP2 inhibits intron retention.

The current manuscript re-examines the role of MeCP2 in splicing using an advanced bioinformatic approach. More specifically, the authors use methods that allow them to tease apart the contribution of DNA methylation per se, from the contribution of MeCP2 itself. Their conclusion is that, once the confounders are removed, there is little evidence to suggest that MeCP2 actually regulates splicing extensively.

The paper is well written and argued. The figures are clear and support the conclusion. I liked the discussion, which is lucid and insightful.

I cannot judge the adequacy of the machine learning approaches used, and I have no technical comments as I am not a bioinformatician.

My main question is biological: I understand using the TKO cells as they are devoid of DNA methylation, but aren’t they also quite low for MeCP2 expression? How can we rule out that there is no effect on splicing because MeCP2 is absent or low? If the authors could show the FPKMs for MeCP2 in ES cells, in comparison to the cells used in the Nat Comm paper (promyelocytes/granulocytes), that would help interpretation.

Reviewer #3: In the current manuscript Chhatbar et al test the effect of a DNA methylation-binding protein, MeCP2, on alternative splicing. The manuscript is based on re-analyses of published data in diverse model systems: ESCs, developing neurons, and hematopoietic differentiation. The authors conclude that MeCP2 has a minimal effect on splicing, and advocate for caution when drawing conclusions that arise from observational high throughput studies. Overall, the manuscript is well-written, easy to follow and the statistical approaches seem appropriate. Nevertheless, there are caveats in this study that the authors need to address before this manuscript is made available for publication.

In the introduction, the authors claim: “Here we revisit the evidence for this phenomenon, leveraging recent, high-quality data sets which track transcriptomic changes in experiments specifically designed to reflect changes in MeCP2 activity.”

However, many of the datasets used in the study are not “high quality”. For example, some of the ESC data (both wt and KO) from the Ast lab are sequenced at a depth of 40M reads (SE, 50bp) which would be insufficient or barely sufficient for proper AS data analyses. Similarly, the brain neuron RNA-seq data is sequenced at 30M or less (50bp SE). Vip neuron data are sequenced at 20M, which is fairly low and insufficient for any meaningful AS analysis. The authors should provide sequencing depth for each sample (ideally in millions of mapped reads) in Table 1.

It should also be explained better why the selected samples were chosen and how their suitability for AS analyses was determined. As sequencing depth plays a major role in AS detection, a few sentences on the role and effects of depth in the current analyses would be most welcome.

The y axes in figures S1B and S1D need to be adjusted. I don’t think it is necessary to use the same scale for differential gene expression and differential splicing. I understand that the authors want to show that the number of differential splicing events is minimal when compared to the number of differentially expressed genes, however it would be much better to represent the splicing data on a separate scale (ie. 1-100 or something like that).

It would be beneficial if the authors could produce a figure with MeCP2 expression levels across different tissues / samples used in the current manuscript.

Finally, as this study only discusses the global impact of MeCP2 on AS and does not try to confirm or refute potential locus specific AS events impacted by MeCP2 binding, the title needs to be adjusted. Something like: “Quantitative analysis questions the role of MeCP2 as a genome-wide (global?) regulator of alternative splicing” would be a better fit.

**Have all data underlying the figures and results presented in the manuscript been provided?**

Reviewer #1: None

Reviewer #2: Yes

Reviewer #3: Yes

PLOS authors have the option to publish the peer review history of their article (what does this mean?). If published, this will include your full peer review and any attached files.

Reviewer #1: No

Reviewer #2: No

Reviewer #3: No

---

## [Decision Letter · Decision Letter 1]

28 Aug 2020

Dear Dr Sanguinetti,

We are pleased to inform you that your manuscript entitled "Quantitative analysis questions the role of MeCP2 as a global regulator of alternative splicing" has been editorially accepted for publication in PLOS Genetics. Congratulations!

Yours sincerely,

Dirk Schübeler

Associate Editor

PLOS Genetics

John Greally

Section Editor: Epigenetics

PLOS Genetics

Comments from the reviewers (if applicable):

Reviewer's Responses to Questions

**Comments to the Authors:**

Reviewer #1: The authors have addressed all my concerns. The inclusion of new datasets now better support their initial conclusions. I furthermore appreciate the change of the title.

Reviewer #2: My concerns on the first version were rather minimal and have been addressed in the revision

Reviewer #3: The authors have successfully addressed all of the reviewers' concerns. I have no further queries.

**Have all data underlying the figures and results presented in the manuscript been provided?**

Reviewer #1: Yes

Reviewer #2: Yes

Reviewer #3: Yes

PLOS authors have the option to publish the peer review history of their article (what does this mean?). If published, this will include your full peer review and any attached files.

Reviewer #1: No

Reviewer #2: No

Reviewer #3: No

**Data Deposition**

http://datadryad.org/submit?journalID=pgenetics&manu=PGENETICS-D-20-00858R1

**Press Queries**

---

## [Editor Report · Acceptance letter]

2 Oct 2020

PGENETICS-D-20-00858R1 

Quantitative analysis questions the role of MeCP2 as a global regulator of alternative splicing 

Dear Dr Sanguinetti, 

We are pleased to inform you that your manuscript entitled "Quantitative analysis questions the role of MeCP2 as a global regulator of alternative splicing" has been formally accepted for publication in PLOS Genetics! Your manuscript is now with our production department and you will be notified of the publication date in due course.

With kind regards,

Matt Lyles

PLOS Genetics

On behalf of:
